# A New Heart-Cutting Method for a Multiplex Quantitative Analysis of Steroid Hormones in Plasma Using 2D-LC/MS/MS Technique

**DOI:** 10.3390/molecules28031379

**Published:** 2023-02-01

**Authors:** Marcela Kotasova, Ondrej Lacina, Drahomira Springer, Jan Sevcik, Tomas Brutvan, Jana Jezkova, Tomas Zima

**Affiliations:** 1Laboratory Diagnostics and Institute of Medical Biochemistry, The First Faculty of Medicine, General University Hospital, Charles University, 12800 Prague, Czech Republic; 2Department of Food Analysis and Nutrition, University of Chemistry and Technology, 16628 Prague, Czech Republic; 33rd Department of Internal Medicine—Endocrinology and Metabolism, The First Faculty of Medicine, General University Hospital, Charles University, 12800 Prague, Czech Republic

**Keywords:** two dimensional-liquid chromatography with tandem mass spectrometry, 2D-LC/MS/MS, multiplexed analysis of steroid hormones, QuEChERS method

## Abstract

The aim of the current research was to develop a simple and rapid mass spectrometry-based assay for the determination of 15 steroid hormones in human plasma in a single run, which would be suitable for a routine practice setting. For this purpose, we designed a procedure based on the 2D-liquid chromatography-tandem mass spectrometry with a minimalistic sample pre-treatment. In our arrangement, the preparation of one sample takes only 10 min and can accommodate 40 samples per hour when tested in series. The following analytical run is 18 min long for all steroid hormones. In addition, we developed an independent analytical run for estradiol, significantly increasing the assay accuracy while taking an additional 10 min to perform an analytical run of a sample. The optimized method was applied to a set of human plasma samples, including chylous. Our results indicate the linearity of the method for all steroid hormones with squared regression coefficients R^2^ ≥ 0.995, within-run and between-run precision (RSD < 6.4%), and an accuracy of 92.9% to 106.2%. The absolute recovery for each analyzed steroid hormone ranged between 101.6% and 116.5%. The method detection limit for 15 steroid hormones ranged between 0.008 nmol/L (2.88 pg/mL) for aldosterone and 0.873 nmol/L (0.252 ng/mL) for DHEA. For all the analytes, the lowest calibration point relative standard deviation was less than 10.8%, indicating a good precision of the assay within the lowest concentration of interest. In conclusion, in this method article, we describe a simple, sensitive, and cost-effective 2D-LC/MS/MS method suitable for the routine analysis of a complex of steroid hormones allowing high analytical specificity and sensitivity despite minimal sample processing and short throughput times.

## 1. Introduction

Steroid hormones play an essential role in many regulatory systems, such as the immune system, stress response, mineral balance, and in the development of sexual characteristics [1,2]. The precise and rapid determination of their concentration is crucial for diagnostics and monitoring of many disorders, such as Cushing’s syndrome, Addison’s disease, and congenital adrenal hyperplasia, as well as personalized treatment. Currently, two different methods are used for the analysis of steroid hormones in both clinical and basic research practice: Immunoassays and tandem mass spectrometry-based methods. Despite several flaws, such as cross-reactivity, high variability between different commercial kits [3,4], and only a “single hormone per assay” approach [5], immuno-based assays are the most frequently used method in clinical practice. This is primarily because of the simplicity and cost effectiveness of the test. The mass spectrometry-based methods offer high sensitivity and specificity of the analysis and enable a metabolomics approach by scoring a panel of molecules in a single run. The largest limitation of mass spectrometry in routine practice is the low-cost effectiveness as well as time-consuming and difficult sample preparation. Biological samples, such as plasma, urine, or saliva, contain many matrix components with ion suppression effect. Their presence in the analyzed sample significantly decreases the sensitivity. Therefore, the matrix components must be removed from the sample prior to analysis by a pre-treatment method, such as liquid-liquid (LLE) or solid-phase extraction (SPE) [3,4,6].

An alternative to extraction-based sample preparation is two-dimensional liquid chromatography (2D-LC). This approach allows a small fraction of a sample to be separated, and the orthogonal selectivity results in a reduction of matrix effects, resulting in an increased sensitivity of the analysis [7]. The combination of two orthogonal phases is similar to two-dimensional liquid chromatography systems, however, it utilizes different separation principles. Such a configuration has already been designed for an MS-based analysis of pesticide residues in food [8].

The presented methodologic approach simplifies the sample preparation to a 10-min-long pre-treatment followed directly by the analysis, thus providing consistently highly sensitive and specific results. Additionally, the analysis of all types of classical steroid maps, including cortisol, aldosterone, and other steroids in a single run, enables more complex clinical assessment and decreases the associated costs in comparison with immune methods with single molecule per run establishment.

This makes the presented method suitable for use in the routine analysis of complex steroid hormones, allowing for high analytical specificity and sensitivity despite minimal sample processing, short throughput times, and reasonable cost-effectivity.

## 2. Results

### 2.1. Method Development

One of the main limitations of LC-MS analysis in routine practice is the difficult and time-consuming sample preparation. Our main goal was to make the analysis as straightforward as possible while keeping the specificity and sensitivity sufficient for clinical and experimental purposes. We employed the QuEChERS method, which consists of technically simple phase partitioning [8,9,10]. This enabled the preparation of a peptide-free concentrated non-polar ACN fraction containing steroids and amphipathic molecules. The interfering phospholipid contaminants were subsequently trapped on the HILIC column and removed before the analytical separation. This step significantly improved the ionization of steroid analytes and furthermore extended the lifetime of the used RP-column and thus decreased the costs of the analyses. Particular steroid analytes were subsequently separated on C18 column and quantitatively analyzed on MS (Figure 1). The exact valve settings are depicted in the Appendix A. The reproducibility of retention time between particular runs was analyzed within 3 months for all QC concentrations (Table 1).

As a result, the combination of QuEChERS-based sample preparation and 2D-LC separation enabled processing single plasma sample and the quantified panel of 15 steroid hormones within 28 min while repeatedly reaching high sensitivity and specificity of the analysis.

### 2.2. Recovery and Matrix Effect

The recovery of 15 steroid hormones and their ISTD and the influence of the matrix on the suppression/enhancement of ionization within the designed 2D-LC/MS/MS were further tested.

The absolute recovery ranged between 101.6 and 116.5% (median 111.6%) and after correction with the internal standards, the relative recovery spanned from 99.9% to 106.1% (median 101.0%). The matrix effect was between 75.6 and 139.5% (median 103.3%), the maximal positive effect was observed for DHEAS. The relative matrix effect was in the range of 94.4–103.9% (median 100.6%). Complete data are shown in Table 2.

The results document that the recovery rate and matrix effect are sufficient for the application of the proposed method in practice.

### 2.3. Precision and Accuracy

Precision and accuracy were performed on all levels of the quality control (QC) samples, which are traceable to the reference material, and were done in ten replicates.

The results show the relative standard deviation (RSD) for all analytes (except of estradiol) at all QC levels within-runs and between-runs varied between 0.9% and 6.4%, and the accuracy varied between 92.9% and 106.2% (data are shown in Table 3).

In order to increase the precision of the analysis specifically for estradiol, the independent analytical run (IAR) was designed. The parameters of this IAR for estradiol were: 1.0% of the within-run precision (while the classic analytic run had 6% for the estradiol), 4.7% of the between-run precision (while the classic analytic run had 17.2% for the estradiol), the within-run accuracy varied between 90.2% and 99.9%, and the between-run accuracy varied between 80.9% and 102.2%.

The inter-day precisions were evaluated by analysis of three replicates of quality control samples during five consecutive days. The accuracy and precision were evaluated from the same quality control samples, and the percent recovery ranged from 94.05% to 106.98% (Table 4).

The sensitivity of the method was considered well-suited for the measurement of the panel of 15 steroid hormones, including a sample of estradiol in plasma prepared by the minimalistic process.

### 2.4. Linearity and MDL

Linear regression was used to construct calibration curves with R^2^ ≥ 0.9991, except for estradiol for the total method, where R^2^ = 0.975 compared to IAR R^2^ = 0.9999. For MDL, the independent analytical run obtained better results, with half of the value of MDL and RSD. The linearity and MDL are presented in Table 5.

The MDL values were found to be less than or equal to 50 pmol/L for all steroids, except for DHEA (873 pmol/L) and estradiol (80 pmol/L). These MDL values were considered suitable for the quantification of each of the analyzed steroid hormones.

## 3. Discussion

Currently, immunoassays are the most frequently used methods for the routine analysis of steroid samples, despite their considerable shortcomings. Alternatively, 2D-LC/MS/MS offers higher specificity and multiplex testing. Several methods using the 2D-LC arrangement for the analysis of the steroid hormones in various sample types have been published. However, these methods have several weak points that significantly limit their use in routine clinical practice.

One of the main limitations of LC-MS analysis in routine practice is the difficult sample preparation. This step is frequently based on time-consuming liquid-liquid extraction with subsequent solvent evaporation. Moreover, due to the non-polar character of the used solvent, the extract also contains some amount of polar phospholipids that frequently interfere with the subsequent MS analyses. Alternatively, solid-phase extraction uses cartridges with different sorbents to clean up the samples [3], but it is far more expensive due to the complicated preparation of the SPE cartridge. When used on a complex biological matrix, it still requires initial pre-treatment of the sample, such as deproteinization or dilution [7,11,12,13]. Further limitations are “only few steroid hormones per sample” analysis and associated high-cost effect.

Our main goal was to make the analysis as straightforward as possible while keeping the specificity and sensitivity sufficient for clinical and experimental purposes. First, we tested the possibility of injecting the crude plasma directly after precipitation with ACN. However, the remaining plasma peptides caused fast deterioration of the separation on HILIC, and the extract was too diluted to achieve a good sensitivity for all analytes. Moreover, the plasma extract contained more than 30% of water, which further limited the amount of sample injected. Second, we tested a golden standard way based on extraction of plasma sample with tert-Butyl methyl ether. The main issue of this process is the very low recovery of DHEAS, which does not pass to the extract due to the presence of thio group in the structure. Contrary to DHEAS, phospholipids are concentrated in the extract and strongly suppress the ionization of analytes during subsequent MS-based detection. In order to overcome the above-mentioned issues, we employed the QuEChERS method [8,9,10] for the sample preparation, followed by the 2D liquid chromatography separation. QuEChERS (from the “Quick, Easy, Cheap, Effective, Rugged and Safe) is an extraction method with a phase partition step, originally designed for the determination of various pesticides. In our QuEChERS arrangement, the plasma sample is directly extracted by ACN with further addition of a high concentration of salts and magnesium sulfate. This causes the plasma protein precipitation and separation of two phases—highly polar salt containing water-based phase and non-polar ACN-based phase containing all required steroid analytes (including the DHEAS), while the protein precipitate is in the interphase. This simple preparation results in a small volume sample with a reasonable concentration of analytes. The phospholipid contaminants in the ACN extract are subsequently trapped on the HILIC column and removed during the first level of chromatographic separation. The particular steroid hormones are separated from residual contaminants (neutral lipids) that may negatively influence the ionization by subsequent second analytical separation on C18 column.

In the presented arrangement, the single-step sample preparation took only 10 min (in series: 40 samples per hour), and the following analysis took an additional 18 min while enabling simultaneous analysis of 15 steroid hormones. In our settings, most of the interfering matrix was held on the extraction column and was washed out as waste after switching valves. Due to the optimized valve settings, we were able to remove the lipid portion from the sample and thus significantly increase the lifetime of the RP-column and thus improve the cost ratio of the analyses. Selected fractions of steroid hormones from the HILIC column were retained in TRAP, and after switching valves washed out to RP-C18 for separation. This procedure significantly shortens the time of the sample preparation and analysis and enables the possibility of multiplex analysis.

A common issue of the LC/MS setup is the negative effect of the matrix on the analysis [14]. This is caused by the fact that the dissolved compounds need to be transformed into the gas phase prior to the detection by MS. During this process, other compounds stemming from the biological matrix will be brought alongside the compounds of interest and may affect the ionization process. Matrix effects can increase the ionization of the analyte (ion enhancement) or, more commonly, decrease its ionization efficiency (ion suppression). These effects can be compensated by the use of stable internal labeled standards within the analysis [14,15]. Most of our steroids were detected in positive ionization mode. Only DHEAS and estradiol were more efficiently ionized in negative mode. The maximal positive absolute matrix effect was observed for DHEAS (137.1%) due to the chromatographic settings and ionization effect.

A number of methods for quantification of each analyte in single [16,17] as well as multiple mode [18,19] have been published. Our 2D-LC/MS/MS method arrangement reaches similar or better results with the monitored steroid hormones and their parameters in comparison with current assays, while all the analytical parameters are sufficient for clinical application.

In summary, the absolute recovery ranged between 101.6 and 116.5%, and the relative recovery ranged between 99.9% and 106.1%. During the within-run test, the results’ precision ranged between 0.8 and 4.4%, and the accuracy 92.9–106.2%. The between-run precision reached 1.4–6.4%, and the accuracy ranged between 97.7 and 105.1% for all analytes at three different concentration levels. The method detection limit (MDL) for 15 steroid hormones ranged from 0.008 (2.88 pg/mL for aldosterone) to 0.873 nmol/L (0.252 ng/mL for DHEA). The only exception was DHEAS with the MDL at a higher concentration of 12.7 nmol/L (4.68 ng/mL). For all the analytes, the lowest calibration point relative standard deviation was lower than 10.8%. In conclusion, the analytical parameters of the proposed 2D-LC/MS/MS-based method are perfectly sufficient for routine analysis of the majority of steroid-related pathogenicity. Our new method is user-friendly and does not require any difficult sample preparations, unlike other common methods. Additionally, the ability to quantitatively score 15 different steroid hormones simultaneously makes this method very flexible for analyzing a variety of disorders. Thus, we consider the proposed method to be a good alternative for steroid hormone analysis.

## 4. Materials and Method

### 4.1. Materials and Reagents

Pre-mixed sets of steroid hormones and their internal standard (link labels) were purchased from Chromsystems Instruments & Chemicals GmbH (Gräfelfing, Germany). The Internal Standard MIX MassChrom^®^ Steroids in a Serum/Plasma (ISTD), 6plus1^®^ Multilevel Serum calibrator set MassChrom^®^ steroid panel 1, 6plus1^®^ Multilevel Serum calibrator set MassChrom^®^ steroid panel 2, MassCheck^®^ Steroid Panel 1 Serum Control, Level I, II, and III; MassCheck^®^ Steroid Panel 2 Serum Control, Level I, II, and III; Tuning Mix^®^ MassChrom^®^ Steroid Panel 1, and Tuning Mix^®^ MassChrom^®^ Steroid Panel 2 were used during the measurements. The individual hormones were certified reference-grade material purchased from Sigma-Aldrich (St. Louis, MO, USA). Charcoal-stripped fetal bovine serum (South America) was obtained from Biosera (Shanghai, PRC).

The LC/MS-grade methanol (MeOH), acetonitrile (ACN), formic acid, ammonium fluoride (NH_4_F), and ammonium acetate were all obtained from Sigma-Aldrich. The chemicals for sample preparation: Magnesium sulfate (p.a., powder (very fine)) and sodium chloride (p.a., ≥99.5%), were also obtained from Sigma-Aldrich. Ultra-pure water was produced in-house using the PURELAB FLEX 3 system (Elga Veolia, High Wycombe, UK). The reagents were used without further purification.

For determining the recovery and matrix effect, several mixtures of all hormones were prepared in MeOH. The final concentration ranges in charcoal-stripped fetal bovine serum after the spike were: 11-deoxycorticosterone (11DCOS) 0.109–7.27 nmol/L, 11-deoxycortisol (11DCOR) 0.624–41.6 nmol/L, 17α-Hydroxyprogesterone (17αOHP) 0.909–60.6 nmol/L, 21-deoxycortisol (21DCOR) 0.208–13.9 nmol/L, Aldosterone (ALDO) 0.120–7.98 nmol/L, Androstenedione (ANDRO) 0.712–47.5 nmol/L, corticosterone (COS) 2.12–141 nmol/L, cortisol 11.6–773 nmol/L, cortisone 1.66–111 nmol/L, Dehydroepiandrosterone (DHEA) 3.89–194, Dehydroepiandrosterone sulfate (DHEAS) 325–15176 nmol/L, Dihydrotestosterone (DHT) 0.427–5.50 nmol/L, estradiol 0.264–17.6 nmol/L, progesterone (PROG) 1.14–76.32 nmol/L, and testosterone (TEST) 0.625–41.6 nmol/L. For sample preparation, solutions of internal standards (ISTD) in acetonitrile (ACN—2 mL in 50 mL ACN) and saturated salt solution were used.

### 4.2. Sample Preparation

The plasma samples were pre-treated by deproteinization. First, 0.5 mL of plasma sample, calibration solution, and QC sample were each mixed with 0.5 mL of ISTD in ACN and immediately vigorously vortexed for 30 s. Subsequently, 0.5 mL of saturated salt solution and 0.2 g magnesium sulfate were added to the sample and mixed steadily for 2 min. Finally, the organic layer containing the analyte was separated from the aqueous part by centrifugation (6 min, 15,000 rpm) and used directly for the analysis. A smaller amount of sample up to 150 μL can be used for samples preparation as long as the ratio of solutions remains the same.

### 4.3. D-LC/MS/MS Instrumentation and Analysis

The system for the chromatographic separation was adapted from the previously described methodology [8] and modified for a versatile analysis of various analytes in human plasma. In the current arrangement, the method enables quantifying a set of 15 steroid hormones. The core arrangement consists of two Agilent 1260 binary pumps and two ten-port valves for 2D-LC operation. The first dimension of chromatographic separation presents a YMC—Triart Diol-HILIC column (100 × 2.1 mm, 3 µm; YMC, Japan), and the second dimension was an In-finityLab Poroshell 120 EC-C18 (3 × 100 mm; 2.7 µm; Agilent Technologies, Santa Clara, CA, USA). The switch between particular columns is provided by a loop consisting of InfinityLab Poroshell 120 EC-C18 column guard (5 × 3 mm, 4 µm; Agilent Technologies, Santa Clara, CA, USA) used as SPE cartridge (TRAP). Both analytical columns were held at 40 °C.

The composition of the mobile phase for separation in the first dimension was (A1): Ammonium acetate (final concentration 25 mM) and formic acid (final concentration 0.2%) in water, and (B1): 100% ACN. In the second dimension, the mobile phase consisted of (A2): Ammonium fluoride (final concentration 0.5 mM NH_4_F) in water, and (B2): 100% MeOH. The gradients used in the columns and changing the time of the valves are shown in Figure 1.

Tables of gradients and other data are mentioned in the Appendix A. The sample’s injection volume was 10 μL, and the total time of analysis was 18 min.

In order to increase the sensitivity of the determination of estradiol, we adjusted the analytical run (IAR) for estradiol independently. The volume of injected sample was 20 μL, and the total time of this estradiol analysis was 10 min. Specific settings of the gradient and valve changes are mentioned in Appendix A.

The analytes were detected using an Agilent 6470 triple quadrupole equipped with divert valve and Agilent JetStream ESI source. The optimized source conditions for the steroid hormone method (estradiol IAR) were as follows: Capillary voltage—positive 3000 V, negative −3800 V (−3000 V), nozzle voltage—positive 0 V, negative −500 V (−600 V), desolvation gas temperature was 250 °C (220 °C), sheath gas temperature was 380 °C (350 °C), desolvation gas flow was 8 L/min, sheath gas flow was 12 L/h. Mass transitions for all analytes are presented in the Appendix A. Analytes were detected in dynamic MRM mode, with variable dwell times in the range from 16.00 to 348.00 ms. The precursor, product ions for each analyte, the corresponding optimal collision energies, and their retention times are also shown in Support Information (Appendix A).

### 4.4. Method Parameters

#### 4.4.1. Recovery

To evaluate the recovery, several concentration levels were prepared. At one concentration level, five pre-extraction spiked samples were prepared with the addition of the mixture of the hormones plus ISTD to charcoal-stripped serum. The solution was then deproteinized with pure ACN. Five post-extraction spiked samples were prepared in the same way, except that an additional mixture of hormones and ISTD were added after the sampling procedure to the upper organic layer. Absolute recovery was calculated by comparing the peak areas of pre-extraction vs. post-extraction spikes, and the relative recovery was calculated by comparing the areas corrected to those of the internal standards (in percent).

#### 4.4.2. Matrix Effect

The absolute matrix effect (ME) was calculated by comparing the peak area of the analyte from spiking into the post-extraction matrix with the peak area from spiking into the solvent matrix (mixture of hormones and ISTD to ACN). In the same way, we compared the areas of the peaks corrected by the internal standards (in percent). A matrix effect of 100% equals the situation when the signal from the serum was not influenced by the matrix properties. We analyzed the matrix’s influence on the suppression or enhancement of ionization under the same conditions as in the recovery measurements.

#### 4.4.3. Linearity

To prepare the calibration standards for panel 1, panel 2, and the blank, a Multilevel Serum calibrator set MassChrom^®^ steroid was used. The calibration standards were prepared in the same way as the corresponding sample. The calibration curves were constructed in different concentration ranges, depending on the analyte. Linear regression was used to construct calibration curves with squared regression coefficients R^2^ ≥ 0.995 and weight 1/x.

#### 4.4.4. Method Detection Limit

The method detection limit (MDL) was used as the parameter of sensitivity. The lowest calibration point was measured in ten replicates, including sample preparation. The MDL was calculated according to the following equation [20]: MDL = t_n−1_ × (RSD/100) × c, when t_n−1_—Student “t” value in 99% confidence level, RSD—relative standard deviation of sample area (%), c—concentration of the sample.

#### 4.4.5. Precision

For precision and accuracy, the quality control (QC) samples were prepared at three different concentrations—low, medium, and high. For each analyte at indicated concentrations, 10 replicates were measured for the determination of precision and accuracy in both within- and between-runs, respectively. The intra-day and inter-day precision were determined by analyzing the QC samples repeatedly on the same and three consecutive days, respectively. Precision parameters were characterized by the percent of relative standard deviation (RSD).

### 4.5. Data Analysis and Statistics

Data analysis was performed using Agilent MassHunter Workstation Software v. 1 (Santa Clara, CA, USA) for LC/MS. Data acquisition was performed and processed in Agilent MassHunter Workstation Software for LC/MS. Peak areas for each compound were normalized to the peak area for the corresponding IS in each sample. Analytical performance parameters were analyzed using Excel (Microsoft, Seattle, WA, USA).

## 5. Conclusions

In this work, we introduced a simple and reliable 2D-LC/MS/MS-based method for the multiplex analysis of 15 steroid hormones in a plasma sample in a single run. The designed method enables direct analysis of a plasma samples after simple protein precipitation, followed by phase partition. The arranged analysis consists of a 10-min-long sample preparation directly followed by an 18-min-long analytical run. The simplicity of the approach, together with good parameters of the analysis, makes the proposed method suitable for use in routine clinical practice.

## Figures and Tables

**Figure 1 molecules-28-01379-f001:**
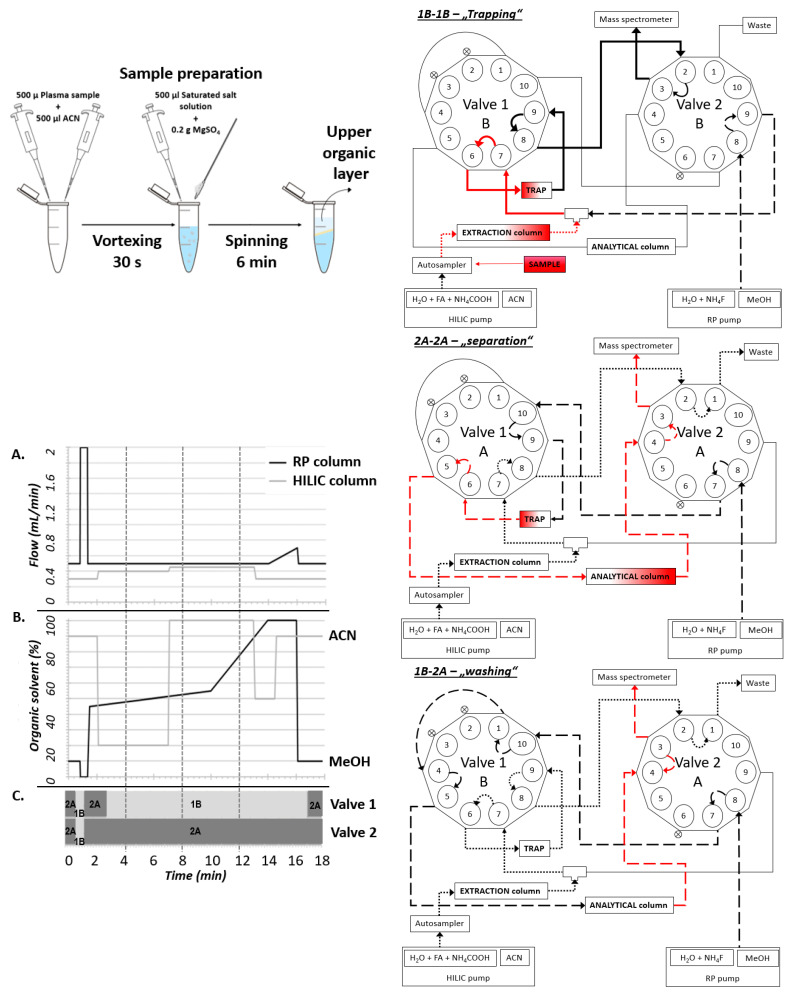
Analysis arrangement: The plasma sample is prepared by the modified QuEChERS-way in a two-step pre-treatment and directly injected into the system (10 min per sample). The target non-polar analytes are separated from the matrix on the extraction column (HILIC) and retained at TRAP (RP column) during the “Trapping” 1B-1B valve setting, while the interfering phospholipids are removed prior to the analytical separation. Subsequently, the analytes are washed out from TRAP, separated in the analytical column (RP), and analyzed in a mass spectrometer during the “Separation” 2A-2A valve setting. Finally, the entire system is reconditioned during the “Washing” 1B-2A valve setting—in total, the run takes 18 min. The flow (**A**) and composition (**B**) of particular mobile phases are depicted with respect to the actual valve setting (**C**) during the run.

**Table 1 molecules-28-01379-t001:** The reproducibility of retention time during between-run measurements for all QC concentrations (3 moths) is given in relative standard deviation % (RSD%).

Analyte	RSD (%)	Analyte	RSD (%)	Analyte	RSD (%)
11DCOS	2.01	Cortisone	1.93	DHEAS	2.81
21DCOR	1.96	11DCOR	0.64	DHT	0.36
ALDO	1.76	17αOHP	0.45	Estradiol	0.89
COS	1.97	ANDRO	0.89	PROG	0.29
Cortisol	2.02	DHEA	0.50	TEST	0.63

**Table 2 molecules-28-01379-t002:** Recovery (R) and matrix effect (ME) of particular steroid hormones in percentage with the relative standard deviation (RSD) analyses by 2D-LC/MS/MS.

Analytes	Absolute (%)				Relative (%)			
	ME	RDS	R	RDS	ME	RDS	R	RDS
11DCOS	100	1.1	115	1.9	100	0.9	101	1.0
11DCOR	105	4.6	113	2.1	102	4.7	100	1.4
17αOHP	105	3.8	115	2.3	94.4	1.3	101	1.4
21DCOR	101	4.5	117	1.6	97.8	1.3	102	2.7
ALDO	105	5.4	104	3.2	101	1.0	102	1.6
ANDRO	106	2.0	114	1.6	103	1.9	102	1.5
COS	102	1.6	112	1.4	99.2	1.3	101	1.8
Cortisol	105	1.7	108	1.7	101	1.5	100	1.4
Cortisone	105	1.4	111	1.2	101	1.3	99.9	1.4
DHEA	104	2.8	108	1.2	104	4.2	106	1.3
DHEAS	137	7.3	115	8.8	100	2.0	102	1.8
DHT	75.6	1.7	105	3.7	97.1	0.7	100	2.9
Estradiol	89.0	4.7	108	4.7	101	1.4	104	1.2
PROG	100	1.9	113	3.1	98.3	0.9	101	0.5
TEST	106	2.9	112	2.0	102	3.3	100	1.7

**Table 3 molecules-28-01379-t003:** Within-run and between-run precision, accuracy for each analyte. *n* = 10.

Analytes	Within-Run Precision			Between-Run Precision			Within-Run Accuracy			Between-Run Accuracy		
	(RSD%)			(RSD%)			(%)			(%)		
	Low	Med.	High	Low	Med.	High	Low	Med.	High	Low	Med.	High
11DCOS	2.5	1.6	2.0	4.1	1.9	1.9	103.5	98.8	97.8	102.2	101.8	100.7
11DCOR	3.3	3.1	1.6	3.5	4.2	1.4	100.3	100.3	100.0	99.5	99.0	99.2
17αOHP	1.8	1.7	0.8	3.8	2.3	2.1	101.0	94.8	93.9	105.1	98.1	98.8
21DCOR	2.5	2.9	1.7	3.3	1.9	2.5	98.1	97.9	99.4	99.6	101.5	100.2
ALDO	3.0	2.4	1.8	3.0	1.8	1.9	99.5	101.5	99.3	100.0	99.4	100.3
ANDRO	0.9	1.2	1.2	4.1	1.4	2.5	106.2	98.1	98.5	100.3	97.7	99.9
COS	3.3	3.0	2.7	4.2	6.2	2.3	96.0	92.9	98.0	101.3	99.6	98.6
Cortisol	1.4	1.4	0.9	1.9	2.3	1.8	100.0	99.4	101.3	99.9	98.7	98.9
Cortisone	2.6	1.2	1.0	1.9	2.7	1.9	100.7	103.7	101.8	99.0	102.6	100.9
DHEA	4.0	3.7	4.4	5.3	4.0	2.9	104.0	99.1	98.3	102.1	100.8	102.4
DHEAS	1.6	1.8	1.2	3.8	3.2	3.1	100.4	97.7	100.1	100.0	99.9	104.3
DHT	2.3	3.2	3.1	6.4	6.4	1.8	104.2	100.6	99.8	98.6	101.3	98.2
Estradiol	6.0	9.3	4.3	17.2	10.5	4.5	90.2	98.6	101.9	80.9	93.7	101.5
Estradiol IAR	1.0	1.9	2.0	4.7	5.0	4.5	99.9	96.9	97.7	102.2	98.2	99.2
PROG	1.8	1.8	1.7	3.4	3.5	2.5	98.5	96.6	98.8	100.4	98.3	101.3
TEST	2.1	1.7	1.4	2.4	2.2	2.7	100.5	98.8	99.1	99.5	99.7	101.7

**Table 4 molecules-28-01379-t004:** Inter-day stability of QC at three levels (low, medium, and high) for 5 consecutive days indicated as percentage of precision and accuracy RSD.

Analyte	Low		Med		High	
	Precision RSD (%)	Accuracy RSD (%)	Precision RSD (%)	Accuracy RSD (%)	Precision RSD (%)	Accuracy RSD (%)
11DCOS	1.50	99.32	1.32	98.72	1.39	99.10
21DCOR	3.80	102.57	2.00	101.13	2.00	100.34
ALDO	2.53	100.78	3.02	97.09	1.31	98.80
COR	1.97	103.36	1.73	99.08	1.21	98.60
Cortisol	1.39	100.10	1.22	97.75	1.19	98.49
Cortisone	0.94	99.41	1.28	99.54	0.80	100.31
11DCOR	3.56	103.47	1.96	101.79	1.92	100.61
17αOHP	4.14	106.98	2.69	98.64	1.21	97.46
ANDRO	4.19	97.98	1.02	97.59	2.18	100.05
DHEA	3.74	101.58	4.49	96.84	5.04	103.54
DHEAS	4.07	99.72	2.65	101.69	4.59	104.31
DHT	3.35	102.98	3.84	101.27	0.73	99.00
Estradiol	3.79	98.79	7.32	94.05	4.67	96.98
PROG	4.64	99.55	4.30	99.49	3.10	101.72
TEST	2.72	98.62	2.44	99.92	3.69	101.58

**Table 5 molecules-28-01379-t005:** Linearity and MDL for all steroid hormones.

Analytes	Range (nmol/L)	Slope	Intercept	R^2^	MDL (nmol/L)	MDL (ng/L)	RSD (%)
11DCOS	0.154–8.91	1.679	−0.030	0.9999	0.009	2.97	2.1
11DCOR	0.312–43.6	0.064	0.000	0.9999	0.022	7.62	2.5
17αOHP	0.315–66.2	0.750	−0.040	0.9996	0.046	15.2	5.1
21DCOR	0.188–13.9	0.574	0.012	0.9999	0.030	10.4	5.7
ALDO	0.075–8.00	1.139	−0.008	0.9999	0.008	2.88	3.7
ANDRO	0.698–46.5	0.357	−0.042	0.9998	0.022	6.30	1.1
COS	1.54–141	0.128	−0.026	1.0000	0.088	30.5	2.0
Cortisol	28.0–785	0.034	−0.145	0.9993	0.241	87.4	0.3
Cortisone	2.88–111	0.141	0.006	1.0000	0.054	19.5	0.7
DHEA	3.62–201	0.156	−0.322	0.9992	0.873	252	8.5
DHEAS	317.0–15895	0.026	−3.564	0.9991	12.7	4680	1.4
DHT	0.189–4.69	0.197	0.020	0.9996	0.057	16.6	10.8
Estradiol	0.158–19.6	0.722	0.008	0.9975	0.080	21.8	17.9
Estradiol IAR	0.158–19.6	0.731	−0.021	0.9999	0.037	10.1	8.2
PROG	0.518–79.6	0.218	−0.004	1.0000	0.023	7.23	1.6
TEST	0.173–42.3	0.157	−0.002	1.0000	0.014	4.04	2.8

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
