# Peer review of "A New Heart-Cutting Method for a Multiplex Quantitative Analysis of Steroid Hormones in Plasma Using 2D-LC/MS/MS Technique"

_molecules, 2023, doi:10.3390/molecules28031379_

Round 1

Reviewer 1 Report

The manuscript presents the heart-cutting 2D-LC/MS/MS method with a related sample preparation procedure for the analysis 15 steroid hormones in the plasma sample.

I suggest changing the title. Multiplex is not so informative, and the heart-cutting 2D-LC/MS/MS expression should be used instead of 2D-LC/MS/MS. I think the application of 2D is not a user-friendly approach.

The discussion part should be more comprehensive. The obtained validation parameters should be compared with earlier published methods.

The validation parameters should be detailed:

-          The reproducibility of retention time is missing.

-          The inter- and intra-day stability assays are missing.

The investigation of trapping efficiency is missing.

I have the following notes:

-          Figure 1. caption: A and B are not labelled. In the case of valve positions, naming the given cycles helps to understand their aims, for example, trapping, equilibrating, measuring, and washing. I think adding the valve positions figure to figures A and B with the same scales also helps to overview.

-          Figure 1.: The figure should be edited (18 is not visible (A). I suggest using the time up to 18 min in the case of B.

-          S Figure A.7: Mass spectrometer instead of “Mass spectrometr”.

-          Table 1 should be edited because of the name of the analytes.

-          lines 211-212: “The matrix effect was between 75.6 - 139.5% (median 103.3%); the maximal positive effect was observed for DHEAS, which was caused by the chromatographic settings.” The matrix effect is also caused during the ionization process.

-          line 257: “R2 = 0.09752” should be corrected to 0.9975

 I think the weakest point of this 2D method is no flow on the analytical column in the valve 1B-1B (trapping) positions. The other is when the HILIC and RP effluent with the sum of 2.3 mL/min flow rate is introduced into the MS during the trapping procedure. Overall, the submitted manuscript provides novelty, but several issues should be clarified.

Author Response

Dear reviewer,

First of all, we would like to thank you for your effort to keep a high scientific soundness of published articles – we do appreciate your attitude and we are grateful for your valuable and professional comments.

  1. I suggest changing the title. Multiplex is not so informative, and the heart-cutting 2D-LC/MS/MS expression should be used instead of 2D-LC/MS/MS. I think the application of 2D is not a user-friendly approach.

Thank you. We do agree with the suggestion to change the title – it reflects the reality more precisely.

As for the “user friendly” - once the method was optimized and set up, we were able to process many samples without any problems. Results of QCs and mainly the responses of internal standards are stable and reproducible. With this approach, we can remove most of interfering co-extracts and get very good sensitivity without a need for evaporation, dissolving and other sample manipulations.

  1. The discussion part should be more comprehensive. The obtained validation parameters should be compared with earlier published methods.

We extended the discussion for a short paragraph mentioning the comparison of analytical parameters of our methods with some of the previously published ones. We did not do any direct experiments focused on the comparison – there is too many parameters, including the sample collection, type of the plastic etc. that should have been taken into the consideration in case of responsibly attitude of comparing the various methods with the one described in our proposal.

  1. The validation parameters should be detailed.

We have added the requested results for the reproducibility of retention time, and the inter-day stability assays. Please see the revised version of manuscript. The intra-day stability is reflected by “within run” accuracy in table 3.

  1. The investigation of trapping efficiency is missing.

The trapping efficiency was tested during the method optimization. We monitored target analytes in the effluent form the trapping column during the trapping period. Since all steroids eluted form the HILIC column in a void volume, the zone is narrow and the trapping period can be short. There was no signal of even the most polar steroids (e.g. aldosterone, DHEAS) from the trapping column.

  1. We have corrected all the indicated mistakes in text and figures.

  1. I think the weakest point of this 2D method is no flow on the analytical column in the valve 1B-1B (trapping) positions. The other is when the HILIC and RP effluent with the sum of 2.3 mL/min flow rate is introduced into the MS during the trapping procedure.

To our knowledge, the no flow through the column is not a big issue. Retention time and responses were stable and reproducible, also the column lifetime is several thousands of injections, which is, to our knowledge, acceptable. Since the analytical column is not UHPLC, sub 2µm, the pressure drops are not huge.

The efluent goes through the built-in divert valve into a waste during the trapping period. We agree that the flowrate of the 2.3 ml/min would be too much for the ESI source, moreover it could cause a quick contamination of the source.

We would be more than happy to receive any other constructive criticism, suggestions and comments which will help to improve the proposal. Hence, we kindly ask you to review the new revised version and consider your further decision.

Thank you very much

Reviewer 2 Report

I read this manuscript with great interest. Although the two-dimensional LC-MS quantitative method of 15 steroid hormones in plasma has been established, there is no specific application, so I think the manuscript is not innovative. Moreover, the method used is a very mature two-dimensional liquid phase method, which is a two-dimensional chromatography technology developed by Angilent Company. The author only uses this technology, without great innovation. Therefore, I suggest rejecting this MS.

In addition, the author had better give the chemical structures of these 15 steroid hormones, so that readers can clearly know their structural differences.

There are no major shortcomings in methodology validation. From the text, the author also referred to the previously reported methods. Therefore, this article is not innovative.

The figures 1 and 2 in the article are too rough. It needs to be carefully improved.

Author Response

Dear reviewer,

First of all, we would like to thank you for your effort to keep a high scientific soundness of published articles – we do appreciate your attitude.

However, we dare to disagree with your professional conclusions of the proposal as a “non-innovative”. The described methodology directly reflects very certain needs of clinicians – here represented by co-authors. Their research intentions in the endocrinology are limited by the analytical parameters (mainly detection limit, sensitivity and versatility) of the most frequently used immuno-based assays. We focused on the mass-spectrometry as a promising alternative. The main aim of the proposed methodology was to simplify the sample preparation and analysis as much as possible. To achieve required simplicity while keeping high analytical quality we tested various combination of already described approaches. Indeed, each of adapted approaches (quechers, 2D liquid chromatography and MS analysis) was described previously and cited properly in this proposal. We feel that uniqueness of our assay design is a successful combination of particular advantages of particular approaches resulting in: 1. super simple sample preparation – adaptation of quechers with slight modification enables us to prepare single plasma sample within 10 minutes; 2. 2D-LC setting minimises the negative matrix effect, enables to score panel of analytes in a single run – here we present the panel of 15 steroid hormones, and increase the cost-effectivity of the process (with all respect to the basic research, the utilisation of proposed method in routine practise reflects the cost-effectivity too); 3. MS-based analysis enables to score very low concentration of analytes; 4. The proposed methodical approach is versatile enabling widening of analysis to other fields – this is a pilot methodical article. Currently we are finishing the modification of proposed methodical approach for analysis of panel of steroid hormones in saliva sample; validation the proposed method for dexamethasone test for diagnostics purposes (Cushing syndrome); and quantification of free fraction of thyroid hormones.

As far as we know, none assay setting comparable to the one presented in the proposal, has been published yet. We truly believe that these facts prove the uniqueness and high potential of the designed method.

Please find the revised version of the proposal enclosed in the system.

We would be more than happy to receive any constructive criticism, suggestions and comments which will help to improve the proposal. Hence, we kindly ask you to re-consider your decision.

Round 2

Reviewer 1 Report

The authors have made the correction. Thus, the manuscript can be published in its current form.

Reviewer 2 Report

The authors have addressed the issues raised by the reviewers. In addition, the author has carefully modified, it can be accepted.